# Risk Factors for Postural and Functional Balance Impairment in Patients with Chronic Obstructive Pulmonary Disease

**DOI:** 10.3390/jcm9020609

**Published:** 2020-02-24

**Authors:** Jaekwan K. Park, Nicolaas E. P. Deutz, Clayton L. Cruthirds, Sarah K. Kirschner, Hangue Park, Michael L. Madigan, Mariëlle P. K. J. Engelen

**Affiliations:** 1Center for Translational Research in Aging & Longevity, Department of Health and Kinesiology, Texas A&M University, College Station, TX 77843, USA; jk.park@ctral.org (J.K.P.); nep.deutz@ctral.org (N.E.P.D.); cl.cruthirds@ctral.org (C.L.C.); sk.kirschner@ctral.org (S.K.K.); 2Department of Electrical and Computer Engineering, Texas A&M University, College Station, TX 77843, USA; hangue.park@tamu.edu; 3Department of Industrial and Systems Engineering, Virginia Tech, Blacksburg, VA 24061, USA; mimadiga@vt.edu

**Keywords:** COPD, postural sway, Berg Balance Scale, functional balance

## Abstract

Reduced balance function has been observed during balance challenging conditions in the chronic obstructive pulmonary disease (COPD) population and is associated with an increased risk of falls. This study aimed to examine postural balance during quiet standing with eyes open and functional balance in a heterogeneous group of COPD and non-COPD (control) subjects, and to identify risk factors underlying balance impairment using a large panel of methods. In COPD and control subjects, who were mostly overweight and sedentary, postural and functional balance were assessed using center-of-pressure displacement in anterior-posterior (AP) and medio-lateral (ML) directions, and the Berg Balance Scale (BBS), respectively. COPD showed 23% greater AP sway velocity (*p* = 0.049). The presence of oxygen therapy, fat mass, reduced neurocognitive function, and the presence of (pre)diabetes explained 71% of the variation in postural balance in COPD. Transcutaneous oxygen saturation, a history of exacerbation, and gait speed explained 83% of the variation in functional balance in COPD. Neurocognitive dysfunction was the main risk factor for postural balance impairment in the control group. This suggests that specific phenotypes of COPD patients can be identified based on their type of balance impairment.

## 1. Introduction

Chronic obstructive pulmonary disease (COPD) is a heterogeneous disease with many extrapulmonary manifestations. Impaired balance function is frequently found in these patients [1] and is associated with functional problems in daily life and an elevated fall risk [2]. Preserving balance function in COPD patients is therefore of critical importance to sustain their quality of life and reduce the risk of falls [2,3]. Hence, the American Thoracic Society/European Respiratory Society guidelines recommended that balance function needs to be assessed after pulmonary rehabilitation in COPD patients [1].

Balance function in COPD has predominantly been assessed using “performance or activity” based measurements (e.g., Berg Balance Scale (BBS), Balance Evaluation System Test (BESTest), timed up and go test, Tinetti scale) [4] and balance confidence questionnaires (e.g., Activities-Specific Balance Confidence (ABC) Scale), however, several limitations of these indirect measurements of (functional) balance have been reported [5,6]. While impaired functional balance has often been observed, a generalization of findings is complicated due to variation in disease characteristics among COPD patients, different variables being examined, and different tests being used. Moreover, commonly used functional balance tests have their own limitations, such as ceiling and floor effects [7] and the amount of minimal detectable changes [8]. 

The postural balance of quiet standing is a direct and objective measure of balance function and has mostly been assessed in the general population [9] and in participants with motor impairment disease (e.g., Parkinson’s, stroke) using force plates. Research measuring postural balance in COPD is still limited and is only conducted during balance challenging conditions. Postural balance was found to be disturbed in a small group of COPD patients in the medio-lateral direction in response to upper limb exercise [10], as well as in the antero-posterior direction when standing on an unstable support surface without vision [11]. Force plate or balance platforms are able to detect small changes in balance in clinical conditions [12], because postural sway is influenced by multiple factors, including sensorimotor function (e.g., afferent signaling as sensory feedback, sensory integration in the central nervous system, and the efferent signaling as a motor command to muscle [12]).

Previous studies in COPD focused mainly on one or more risk factors of balance impairment including age, lung health (e.g., severity of lung disease, COPD phenotype [13], the presence of exacerbations [14]), muscle mass [13], muscle strength, physical activity level [15], mobility, and usage of oxygen [16]. No research has been done examining neurocognitive function and the presence of comorbidities (including the presence of (pre)diabetes) as potential risk factors of balance impairment in COPD.

The present study examined whether, besides functional balance (e.g., BBS), postural balance (e.g., postural sway) is also disturbed during more natural, less challenging conditions in COPD patients with eyes open and during quiet standing as compared to non-COPD subjects. Furthermore, potential risk factors underlying functional and postural balance impairment were evaluated using a large panel of comprehensive methods as body composition, gait speed, muscle function, physical activity level, markers of lung health, neurocognitive function, the presence of comorbidities (e.g., (pre)diabetes), and blood markers were studied in each subject. This detailed insight into the risk factors for balance impairment in COPD patients will provide the mechanistic basis to consider postural and/or functional balance as a therapeutic and preventative target(s) to reduce their risk of falls and improve their quality of life.

## 2. Materials and methods

### 2.1. Subjects

We recruited COPD and healthy control subjects from the MEDIT (MEtabolism of Disease with Isotope Tracers) trial, a large controlled trial in healthy and diseased subjects. Patients with clinically stable COPD under the routine control of the pulmonary clinics (COPD) and subjects without COPD (control) were recruited. In total, 79 subjects were assessed for eligibility, 34 COPD, and 22 control subjects were included for data analysis (Figure 1). The inclusion criteria for both groups were age of 55 to 85 years old, and the ability to walk, sit, and stand independently. COPD subjects were classified as moderate to very severe airflow obstruction (GOLD stage II–IV). All COPD patients were in a clinically stable condition and were not suffering from a respiratory tract infection or exacerbation of their disease at least 4 weeks prior to the study. The control group was matched for age on group level by excluding subjects > 80 years old. The exclusion criteria were major neurological conditions that might affect postural sway (e.g., stroke history, Parkinson’s disease), the presence of an acute illness, a metabolically unstable chronic illness, fever within three days prior to the study day, pre-existent untreated metabolic or renal disease, malignancy, recent surgery, and use of systemic corticosteroids one month prior to the study. A medical history, including the number of exacerbation and medication uses, was assessed as part of the screening process. Sixty-six percent of COPD patients were using bronchodilator medication and sixteen percent inhalation corticosteroids. 

Written informed consent was obtained from all subjects before any measurements were performed. The study was conducted in accordance with the Declaration of Helsinki, and the protocol was approved by the Institutional Review Board of Texas A&M University and was registered on ClinicalTrials.gov (NCT03327181).

### 2.2. Anthropometrics, Body Composition, and Lung Function

All study procedures were identical in both groups and the study day lasted approximately four hours. Body weight and height were measured by a digital beam scale and stadiometer, respectively. Blood pressure was measured on the upper arm after a 5-min rest on chair. Anthropometric and body composition were measured to obtain the body mass index (BMI, kg/m^2^), fat free mass index, fat mass index, and the appendicular skeletal muscle index. Whole body, trunk and extremity (arms and legs) fat mass, and fat-free mass were obtained from all subjects while in a supine position by dual-energy X-ray absorptiometry (Hologic QDR 4500/Version 12.7.3.1 (Bedford, MA)). Spirometry was performed using a hand-held device (Microloop Peak flow Meter, CareFusion, San Diego, CA). Additionally, maximal expiratory and inspiratory pressure were assessed by a hand-held device (Micro Respiratory Pressure Meter). Transcutaneous oxygen saturation was measured using pulse oximetry.

### 2.3. Postural Sway Measurement using the Center-of-Pressure

Center-of-pressure (CoP) displacement data were recorded by a force platform (Advanced Mechanical Technology, Inc, Watertown, MA) over 30 s. Three trials of quiet standing were performed according to standardized verbal instructions given to the subject (e.g., ready go and relax). The subjects were asked to stand barefoot on the force platform with arms parallel to the body, not to talk, and to stand as still as possible for 30 s to measure the postural sway of quiet standing. Subjects were instructed to open their eyes and were allowed to stare at a target on the wall, which was approximately two meters away. To minimize the variation of area in base of support, a designated distance between feet was given [17]. Resting time was allowed between the trials, as needed. A safety harness was provided to prevent potential falls (Appendix A: Picture of method for the force platform). 

CoP data were recorded at 100 Hz sampling frequency, as suggested as a reliable measurement for static posture [18]. Mean sway velocity was calculated by total displacement divided by time in both anterior-posterior (AP) and medio-lateral (ML) directions [19]. The Pythagorean theorem formula was used to calculate displacements in the AP-ML (anterior-posterior and medio-lateral combined) direction. The sway area was calculated as 95% confidence ellipse by a principal component analysis using Matlab code (open-access public repository, Figshare (http://dx.doi.org/10.6084/m9.figshare.1126648) [20]).

### 2.4. Functional Balance Measurement Using the Berg Balance Scale (BBS)

Performance-oriented and comprehensive balance functions were evaluated via BBS [21]. BBS has been reported and used widely as a ‘gold standard’ for clinically assessing static and dynamic balance function because of its documented reliability and validity [22]. A maximum score of 56 indicates a good balance function. The same research staff assisted with the test to maintain consistent inter and intra-rater reliability, which has previously been reported to be 0.98 and 0.99, respectively [21]. 

### 2.5. Skeletal Muscle Function and Gait Speed Measurement

Upper and lower limb skeletal muscle function and gait speed were assessed as markers of physical function. Following warming up, peak leg torque during single-leg extension (at 60°/s) was measured using Kin-Com isokinetic dynamometry (Isokinetic International, Chattanooga, TN). Peak handgrip force using dynamometry (Vernier Software and Technology, Beaverton, OR) was used as a marker of maximum handgrip strength. To evaluate mobility performance, four-meter gait speed (4MGS at fast and usual speed) was measured after one practice trial [23]. Subjects were allowed to use their walking aid and/or oxygen if needed. 

### 2.6. Neurocognitive Function Assessments 

Subjects completed the Trail Making Test (TMT) [24] and Stroop color-word tests (SCWTs) [25], which are known to be simple and sensitive in assessing neurocognitive impairment. TMT consists of two subtasks (Part A and part B), and the TMT difference (Time B—Time A) was calculated. SCWTs consists of three subtasks (I, II, and III), and the interference score was calculated [26,27]. 

### 2.7. Physical Activity, COPD Severity Questionnaires, and Comorbidity Assessment 

Self-reported habitual physical activity level was measured by the Physical Activity Scale for the Elderly (PASE) questionnaire. The COPD Assessment Test (CAT) was performed. The Charlson Comorbidity Index (CCI) evaluated self-reported comorbidities and was cross-checked with a medical history and/or a medical chart. Additionally, a history of falls and near falls in the past 12 months was assessed by interview. 

### 2.8. Blood Analysis to Assess Markers of Metabolic/Clinical Health 

Arterialized-venous blood was put in Li-heparinized or EDTA tubes (Becton Dickinson Vacutainer system, Franklin Lakes, New Jersey, USA), immediately put on ice to minimize enzymatic reactions and was centrifuged (4 °C, 3120× *g* for 5 min) to obtain plasma. A part of the plasma was aliquoted into tubes with 0.1 vol of 33% (w/w) trichloroacetic and then vortexed for the denaturation of proteins. Samples were immediately frozen and stored at −80°C until further analysis. Plasma amino acid concentrations of the essential amino acids (EAA), tryptophan (TRP), large neutral amino acids (LNAA), and the branched-chain amino acids (BCAA) were analyzed batch-wise by LC-MS/MS by isotope dilution, as previously reported [28]. High-sensitivity C-reactive protein (hs-CRP), a systemic inflammatory mediator, was measured using a particle enhanced immuno-turbidimetric assay, and fasting glucose concentration was measured using a hexokinase method (Cobas c111, Roche Diagnostics, Mannheim, Germany).

### 2.9. Statistical Analysis

All results were expressed as means ± standard errors (SE). The normality of the data was tested by D’Agostino-Pearson omnibus normality test, or for small group numbers with the Shapiro–Wilk normality test. The Robust regression and Outlier removal (ROUT) test (Q = 1%) were performed to identify outlier. An unpaired Student’s *t*-test was used to compare the groups. When the normality test failed, an unpaired Mann–Whitney test was performed. Pearson’s correlation or Spearman’s correlation coefficients were analyzed according to the result of a normality test to test the association between postural and functional balance and all demographic variables (age, body weight, height, body mass index, physical activity status, and the number of falls), pulmonary function (Forced Expiratory Volume in 1 s, duration of COPD-related symptoms, number of exacerbation, GOLD stage, CAT score, oxygen usage, and transcutaneous oxygen saturation), body composition related variables (e.g., lean mass, fat mass, appendicular skeletal muscle mass, and appendicular skeletal muscle index), CCI, neurocognitive function (TMT difference and Stroop interference), and muscle strength (inspiratory muscle strength, expiratory muscle strength, maximum handgrip, maximal leg extension force, and maximal leg extension force per kg fat-free mass of lower limb). Correlation coefficients were compared between postural and functional balance. Based on the results, we built-up a best fit multiple linear regression model for each postural balance and functional balance in the COPD and control groups. The statistical packages within GraphPad Prism (GraphPad Software, La Jolla, CA, Version 8) and Matlab (The MathWorks, Inc, Natick, MA) were used for data analysis. The significance level was set at α < 0.05 for all analyses.

## 3. Results

### 3.1. General Characteristics

Although no differences were found in age, sex, body weight, height, BMI, or physical activity level between the COPD and control groups, both groups were characterized by being overweight and a sedentary lifestyle. COPD subjects had a higher CCI (*p* < 0.0001; Table 1). The average duration of COPD-related symptoms was 10.8 years, and 20 out of 34 COPD subjects were on long-term oxygen therapy (continuous, as needed, and/or at night; Table 1).

### 3.2. Balance Function

AP mean sway velocity, as a marker of postural balance, was 23% higher in the COPD group (*p* = 0.0496) and there was a tendency towards a higher AP–ML mean sway velocity (*p* = 0.0752). The sway area did not differ between groups (*p* = 0.9786). The BBS, as a marker of functional balance, had a 3% lower score in the COPD group (*p* = 0.0253; Table 2).

### 3.3. Body Composition, Gait Speed, and Muscle and Neurocognitive Functions 

No differences were found in lean mass, fat mass or skeletal muscle mass. Skeletal muscle strength was impaired in the COPD group, as reflected by a 27% lower inspiratory muscle strength (*p* = 0.0007) and an 18% lower maximal leg extension force (*p* = 0.023). Moreover, both usual and fast gait speeds were slower in the COPD group (both *p* < 0.0001; Table 1). The COPD group needed 93% more time to complete tasks measured by the TMT score (*p* = 0.0032), and there was a tendency towards requiring more time to complete tasks in the SCWT (*p* = 0.0628; Table 1).

### 3.4. Plasma Clinical Markers and Amino Acid Concentrations

The plasma concentrations of EAA, TRP, LNAA, and BCAA were not different between the groups. Both the COPD and control group were characterized by elevated plasma glucose levels. The hs-CRP value was higher in the COPD group, indicating an elevated marker of systemic inflammation (*p* = 0.0463; Table 1).

### 3.5. Relationships Between (Postural and Functional) Balance and Age, Body Composition, Muscle, Physical, and Neurocognitive Function, and Disease Severity

Postural balance (Postural sway velocity in AP direction) was significantly associated with functional balance in the control group only (r: −0.44, *p* = 0.037). Furthermore, a significant association was found with:Age in control (r: 0.56, *p* = 0.005) and COPD (r: 0.36, *p* = 0.034).Body composition: whole-body fat mass (r: 0.42, *p* = 0.012) in COPD.Neurocognitive function: Stroop interference in control and COPD (r: 0.72, *p* = 0.0001 and r: 0.59, *p* = 0.0002, respectively).Comorbidity and disease severity: the presence of oxygen therapy (r: 0.3959, *p* = 0.0204) and CCI (r: 0.4274, *p* = 0.0117) in COPD, the presence of (pre)diabetes (subanalysis of CCI) (r: 0.492, *p* = 0.0031) in COPD, and a longer duration of COPD-related symptoms (r: 0.4618, *p* = 0.0068).Plasma markers: fasting glucose concentration in COPD (r: 0.5441, *p* = 0.0259).

Functional balance (BBS). A significant association was found with:
Age in control (r: −0.49, *p* = 0.020).Blood pressure: systolic blood pressure in control (r: −0.54, *p* = 0.008).Skeletal muscle strength: maximum handgrip strength in COPD (r: 0.4594, *p* = 0.0479).Physical function: gait speed (fast) in control (r: 0.57, *p* = 0.005) and COPD (r: 0.68, *p* = 0.001).Physical activity level: PASE in COPD (r: 0.52, *p* = 0.033).Neurocognitive function: Stroop interference in COPD (r: -0.45, *p* = 0.048).Comorbidity and disease severity: a greater number of exacerbations in preceding year (r: −0.82, *p* < 0.0001), the presence of oxygen therapy (r: −0.51, *p* = 0.0254), and a lower transcutaneous oxygen saturation percentage (r: 0.59, *p* = 0.0076), and CAT score (r: −0.45, *p* = 0.02) in COPD.

### 3.6. Multiple Regression Analysis by Postural and Functional Balance 

In COPD, the presence of oxygen therapy, whole-body fat mass, neurocognitive function, and the presence of pre/diabetes explained 71% of the variation in postural balance (Table 3a), whereas transcutaneous oxygen saturation, the number of exacerbations in the preceding year, and gait speed (fast) explained 83% of the variation in the BBS (Table 3b).

In the control group, neurocognitive function explained 50% of the variation in postural balance (Table 4a), whereas systolic blood pressure only explained 29% of the variation in BBS (Table 4b).

## 4. Discussion

Insight into factors contributing to impaired postural and functional balance in COPD is important to reduce the potential risk of falls by designing a tailored treatment program. Our findings indicate that the presence of oxygen therapy, increased whole-body fat mass, and reduced neurocognitive function are risk factors for the impaired postural balance during quiet standing with eyes open in COPD. In addition, transcutaneous oxygen saturation, a history of exacerbation, and gait speed are risk factors for decreased functional balance in COPD.

### 4.1. Postural Sway in COPD: Sway Direction and Sway Area

Our data show an increased sway velocity during quiet standing in COPD in the AP but not in ML direction. Since there are only a few studies performed on postural sway direction in COPD, it has been argued which direction of postural sway is more impaired [29]. Smith et al. reported that COPD patients have reduced postural control in a ML direction [10], which we did not observe. One possible explanation for the discrepancy is a difference in the data analysis procedure. We used the mean sway velocity as a significant factor for postural sway, whereas Smith et al. used sway range and the root mean square. Additionally, the different postural measurement conditions might have impacted the results. Our subjects had their eyes opened during the 30 s of quiet standing measurement period, and a designated feet distance on the firm surface for all subjects was enforced, whereas others, during balance challenging conditions, used a longer measurement time, a hip-width-based feet distance, and a foam surface. Regarding the sway area, a number of studies used different analytical methods on CoP displacement data, such as convex hull or principal component analysis [30], however, these methods are not sufficiently validated in clinical populations, such as COPD. 

### 4.2. Demographics, Body Composition, and Muscle Function and Postural Balance in COPD

No sex difference was found in postural sway (*p* = 0.8799) in either group, in line with previously published data [31]. The reported prevalence of falls and near falls in the preceding year in COPD was 61%. We did not find a significant correlation between the number of falls and the postural sway, however, the self-reported numbers might be inaccurate [32]. 

Our COPD group was overweight/obese (BMI: 30.5 kg/m^2^) and had metabolic-syndrome-related comorbidities, while our control group was also overweight (BMI: 29.5 kg/m^2^), sedentary, and characterized by slightly reduced cognitive function and elevated glucose levels. Although control subjects fulfilled all required inclusion and exclusion criteria, they also had early characteristics of metabolic syndrome. The elevated BMI and sedentary lifestyle of both the control and COPD groups have been observed in our previous studies (COPD [33] as well as in cancer [34]) due to the US population in general getting more obese, independent of the presence of disease [34]. Therefore, we think that our randomly selected control group provides a good representation of the current older population without COPD. Studying physically active controls without the presence of obesity or prediabetes would have likely resulted in a larger difference in balance function between the COPD and control group. 

Previous studies found disturbed postural balance in COPD patients during conditions that challenged balance [10,11]. The present study examined postural balance in COPD patients during a more natural condition with eyes open and during quiet standing. Smaller differences in postural balance between the groups that had been previously observed by others were therefore expected.

Although balance function and overall skeletal muscle strength were impaired in COPD patients, we did not find a significant relationship between measures of limb muscle strength and postural sway, suggesting that the muscle group studied for strength analysis is possibly not relevant to postural sway adjustment. We measured the isokinetic extension of the leg at a speed of 60°/s, and at this speed the maximal voluntary contraction requires predominantly fast-twitch muscle fibers (type II) [35]. Besides the knee, the ankle and hip joints play a large role in postural adjustment [36]. 

### 4.3. Diabetes and Balance Impairment in COPD

In the present study, a significant relationship was found in COPD between postural sway and CCI. Interestingly, the presence of (pre)diabetes and higher fasting glucose were both correlated with increased postural sway. In line with this, postural sway was associated with diabetes-related factors, such as elevated fat mass and blood pressure. These data suggest that the presence of (pre)diabetes may contribute to postural balance impairment in COPD. This could be explained by neurological dysfunction, which is a common complication in diabetic patients [37]. Diabetic neuropathy might lead to impairments of neurological pathways, such as slower motor and sensory nerve conduction, which has been reported in prolonged diabetic conditions [38]. Approximately 50% of diabetics experience polyneuropathy during their lifetime [39], which may lead to balance impairment in these patients. 

Furthermore, the moderate relationship between impaired postural sway and the presence of diabetes-related factors might be explained by (1) a latency of sensory input in postural disturbance relating to three major sensory systems of posture (vestibular, visual, and somatosensory function) [40] and/or (2) a higher amount of posture adjustment [41] with a fine control over a muscle, relating to the neuromuscular system. For example, diabetic patients frequently exhibit an increase of somatosensory deficit [42], such as conduction delay in central and peripheral nervous systems [43]. In addition, polyneuropathy with abnormalities in nerve conduction showed a slower reaction time, a worse static balance, and an increased number of falls. Appenzeller et al. suggested that long-lasting duration of COPD might lead to the breakdown of peripheral myelin and reduced nerve conduction velocity [44]. These complications between diabetes and COPD could contribute to an impaired response of posture adjustment. Our data suggest that the (pre)diabetes phenotype within COPD is at risk for postural balance impairment. The exact mechanisms underlying the link between these two conditions and postural balance impairment deserve further investigation. 

The average glucose concentration was elevated in the COPD and control groups, indicating that both groups were pre-diabetics. The presence of (pre)diabetes, however, did not explain the variation in postural balance in the controls. The number of subjects that were on glucose lowering oral medication was higher in the COPD group (26% vs. 18%). We have not measured the duration of the self-reported comorbidity of diabetes or whether nerve damage was actually (more) present in COPD. This is certainly an area of interest for future research.

### 4.4. Severity of Hypoxemia and Balance Impairment in COPD

Usage of oxygen therapy and decreased transcutaneous oxygen saturation were associated with both postural and functional balance impairment in COPD. Neurocognitive performance was lower in COPD than the control group and the strong association between functional balance and neurocognitive function was particularly present in COPD. A possible explanation is an attention deficit present in COPD patients, particularly in those with (intermittent) hypoxemia [45]. Attention deficit may contribute to longer reaction time in balance adjustment because of the latency in cognition and reaction in postural muscle [46]. Another explanation is that impaired balance is a consequence of hypoxemic cerebral disturbance [47] and/or dysfunction of the sensory reception and integration caused by hypoxia-related neuronal damage [45,48]. Structural change of the brain in COPD (i.e., decreased grey matter) was recently reported [49], which might affect sensory input and motor output processes, especially motor controlling related to balance impairment in COPD. 

### 4.5. Comparison between the Postural and Functional Balance Tests

Functional but not postural balance in COPD is affected by muscle strength and physical performance (gait speed). In agreement, Roig et al. identified gait deficit, besides muscle weakness, nutritional depletion, impaired activities of daily living, and the number of medications as important risk factors for falls in COPD [50]. A low postural balance, however, is associated with a higher duration of COPD symptoms, higher CCI, and more diabetes-related factors (e.g., BMI, blood pressure, fat mass, and plasma glucose concentration). These results indicate that the outcome of the postural and functional balance tests have similar risk factors (e.g., age, hypoxemia) as well as test specific risk factors. The BBS likely represents “functional” aspects of balance, which is particularly present in daily living activities, whereas the postural balance function is more related to biomechanical aspects of the individual’s balance function, which is more relevant when examining the recognition of sensory input and the integration of sensory information.

The advantage of the postural balance test over the BBS is that it has no ceiling or flooring effects. BBS showed a ceiling effect in our population, in line with the previously reported ceiling effect and reliability issues [51]. This might explain why no significant association was found between CoP velocity and the BBS in our study.

The BBS has a maximal score of 56, which was obtained in 52% of the healthy subjects and in 15% of the COPD group. Hence, the BBS is not very sensitive, particularly in control subjects, and the difference in functional balance is likely underestimated due to this ceiling effect.

Previous studies mainly focused on one or more risk factors of balance impairment, including age, lung health [13,14], muscle mass [13], muscle strength, physical activity level [15], limited mobility, and oxygen usage [16]. We extended this by adding cognitive function and the presence of comorbidities (including presence of (pre)diabetes and plasma glucose) to create a large and more comprehensive panel of methods to be used in a single subject. Both factors increased the explained variation in balance in our COPD group.

## 5. Study limitations

Our study had several limitations. First, as of the day of analysis, 79 subjects (age 55–85) completed our (balance) study protocol. The subjects were part of the MEDIT trial, which is an active and still recruiting study of healthy and diseased subjects (see Methods section). Once we started the statistical analysis, we observed that age was not well balanced between the groups. Therefore, we excluded the subjects with age > 80 years old who were all control subjects to better match the mean age of the groups. Second, although we did not find a difference in body weight and height between groups, we assumed that the individual subjects had a difference in height of center-of-mass, which is a known factor affecting postural sway [52]. Therefore, it has been suggested that assessing the height of center-of-mass and CoP simultaneously might improve the accuracy of results for postural balance measurements [53]. Furthermore, not all subjects conducted the BBS measurement, which reduced the power of the correlation analysis between the BBS and postural balance data.

## 6. Conclusions

In the present study, we demonstrated specific phenotypes of COPD patients based on the type of balance impairment. Comorbidity, including (pre)diabetes, the presence of oxygen therapy, increased fat mass, and decreased neurocognitive function, were risk factors for impaired postural balance in COPD. The degree of oxygen desaturation, a history of exacerbations, and physical function were risk factors for impaired functional balance performance. As each balance test reflects unique aspects of balance function and has its own limitations, using both methods in the same patient will increase the sensitivity of detecting balance impairment. Moreover, further research into the mechanisms of postural and functional balance impairment in COPD is needed, and the use of dynamic test conditions with a motion capture system might further characterize disturbances in balance function in COPD. A longitudinal study might reveal whether impaired postural balance and/or functional balance lead to an increased fall incidence in this population so that specifically targeted rehabilitation programs can be developed.

## Figures and Tables

**Figure 1 jcm-09-00609-f001:**
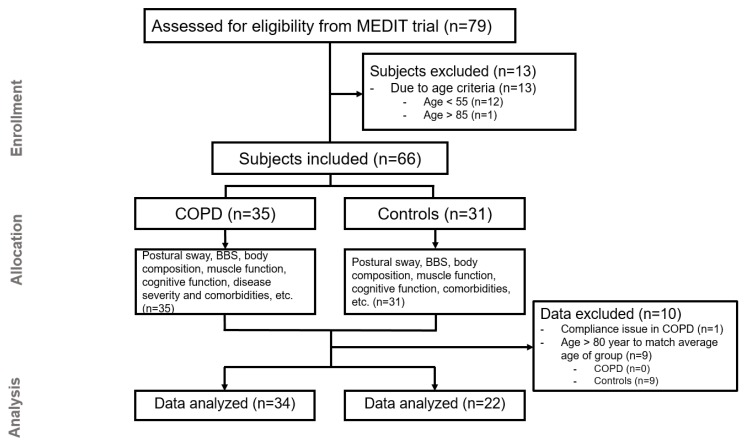
Consort flow diagram of the study.

**Table 1 jcm-09-00609-t001:** Characteristics of the chronic obstructive pulmonary disease (COPD) and control groups.

	Control (*n* = 22)	COPD (*n* = 34)	*p* Value
**General and clinical characteristics of the COPD and Control groups**			
Age (years)	70.44 (1.72)	68.97 (1.36)	0.505
Sex (Male/Female)	11/11	14/20	0.588
Body Weight (kg)	82.36 (2.26)	83.32 (3.47)	0.838
Height (m)	1.67 (0.01)	1.65 (0.01)	0.326
Body Mass Index (kg/m^2^)	29.50 (0.79)	30.53 (1.20)	0.528
Charlson comorbidity index (score)	0.31 (0.12)	2.08 (0.25) **	< 0.0001
Physical activity level (PASE score)	122.0 (18.22)	106.4 (12.46)	0.507
Number of subjects who had a fall and/or near fall history within last 12 month ^1^	N/A	21	-
**Pulmonary function and COPD-related measures**			
FEV1 (% of predicted)	96.91 (2.96)	44.18 (3.13) **	< 0.0001
Transcutaneous oxygen saturation (%)	97.32 (0.33)	95.00 (0.70) *	0.014
Duration of COPD-related symptoms (years)		10.82 (1.11)	
No. of hospitalizations in last year for exacerbation		0.26 (0.10)	
No. of exacerbations in the past year		0.73 (0.24)	
CAT (score)		21.00 (1.26)	
GOLD Stage		2.87 (0.13)	
Dyspnea Scale		2.09 (0.18)	
Oxygen therapy usage (yes/no)	0/22	20/14	
**Body Composition**			
Lean mass (kg)	49.00 (2.27)	48.51 (1.94)	0.871
Lean mass extremities (kg)	20.77 (0.91)	19.41 (0.90)	0.318
Fat mass (kg)	28.56 (1.65)	32.28 (1.91)	0.178
Fat mass index (kg/m^2^)	10.36 (0.68)	11.95 (0.78)	0.163
Fat-free mass index (kg/m^2^) ^2^	19.09 (0.44)	18.46 (0.63)	0.478
Appendicular skeletal muscle index (kg/m^2^) ^3^	7.365 (0.22)	7.053 (0.25)	0.4
Fat % android/gynoid (ratio) ^4^	1.101 (0.05)	1.015 (0.04)	0.205
**Muscle function**			
Inspiratory muscle strength (cmH_2_O)	83.50 (4.27)	60.97 (4.22) **	0.0007
Expiratory muscle strength (cmH_2_O)	100.9 (8.02)	82.71 (5.90)	0.068
Maximal handgrip strength (N)	235.8 (16.58)	203.4 (10.47)	0.087
Maximal leg extension force (N)	257.7 (13.31)	210.5 (13.86)*	0.023
Maximal leg extension force per kg fat-free mass (N/kg)	4.828 (0.20)	4.152 (0.19)*	0.026
**Physical performance**			
Usual gait speed (m/sec)	1.23 (0.04)	0.95 (0.04)**	< 0.0001
Fast gait speed (m/sec)	1.93 (0.07)	1.33 (0.05)**	< 0.0001
Neurocognitive function			
TMT difference (time B–A)	24.61 (2.828)	47.74 (6.370) **	0.0032
Stroop interference (score)	50.95 (4.756)	61.66 (3.924)	0.0628
**Plasma clinical markers & amino acid concentrations**			
EAA (μmol/L)	589.4 (23.53)	588.1 (35.14)	0.975
TRP (μmol/L)	38.33 (1.91)	38.13 (1.84)	0.939
LNAA (μmol/L)	706.8 (25.29)	703.9 (36.29)	0.9476
BCAA (μmol/L)	322.1 (15.50)	329.2 (27.17)	0.818
Glucose concentration (mmol/L)	5.46 (0.129)	5.49 (0.113)	0.6331
Hs-CRP (mg/L)	1.40 (0.265)	3.80 (0.872)*	0.0463

Values are mean ± SE. Statistics are by an unpaired t-test or a Mann–Whitney test when normal distribution test failed. Categorical data were analyzed with the Chi-square test.* = *p* < 0.05; ** = *p* < 0.01. PASE: Physical Activity Scale for Elderly. ^1^ Fall (near fall) history was determined by the standardized question. FEV1: Forced Expiratory Volume in one second. CAT: COPD Assessment Test. COPD: chronic obstructive pulmonary disease. GOLD: Global Initiative for Chronic Obstructive Lung Disease. ^2^ Fat-free mass index = (muscle mass + bone mineral content)/height^2^. ^3^ Appendicular skeletal muscle index = (lean mass legs + lean mass arms)/height^2^. ^4^ Android fat and gynoid fat correspond to central and peripheral fat distribution, respectively. TMT: Trail Making Test. EAA: sum of histidine, isoleucine, leucine, lysine, methionine, phenylalanine, threonine, tryptophan, and valine. LNAA: large neutral amino acids, which is the sum of leucine, phenylalanine and tyrosine. BCAA: the sum of the branched-chain amino acids valine, leucine, and isoleucine. Hs-CRP: High Sensitivity C-Reactive Protein. N/A: data not available.

**Table 2 jcm-09-00609-t002:** Balance function by postural sway and the Berg Balance Scale of the COPD and control groups.

	Control (*n* = 22)	COPD (*n* = 34)	*p* Value
**Postural balance—Postural sway measurement by force platform**			
AP mean sway velocity (cm/s)	0.86 (0.061)	1.11 (0.073) *	0.0496
ML mean sway velocity (cm/s)	0.74 (0.051)	0.86 (0.065)	0.4485
AP–ML mean sway velocity (cm/s)	1.27 (0.084)	1.62 (0.110)	0.0752
Sway area (cm^2^)	3.46 (0.455)	3.41 (0.319)	0.9786
**Functional balance—Performance oriented measure ^†^**			
Berg Balance Scale (score)	54.71 (0.34)	53.11 (0.55) *	0.0253

Values are mean ± SE. Statistics are by unpaired t-test or Mann–Whitney test when normal distribution test failed. * = *p* < 0.05. ^†^
*n*_control_ = 22 and *n*_COPD_ = 29. AP: Anterior-posterior direction (forward-backward). ML: Medio-lateral direction (left-right). AP-ML: AP-ML combined direction.

**Table jcm-09-00609-t003a:** (**a**)

	Coefficients	SE	|t|	*p* Value
Intercept	−0.4461	0.2417	1.845	0.0752
Presence of oxygen therapy (Y/N)	0.4062	0.125	3.249	0.0029
Fat mass (kg)	0.02072	0.00556	3.726	0.0008
Stroop interference (score)	0.009796	0.002222	4.409	0.0001
Presence of diabetes/prediabetes (Y/N)	0.451	0.1332	3.385	0.0021

**Table jcm-09-00609-t003b:** (**b**)

	Coefficients	SE	|t|	*p* Value
Intercept	−5.182	21.23	0.2441	0.8105
Transcutaneous oxygen saturation (%)	0.5577	0.2166	2.575	0.0211
Exacerbation in the last year (number)	−1.165	0.3545	3.286	0.005
Gait speed (fast) (m/s)	3.972	1.395	2.847	0.0122

**Table jcm-09-00609-t004a:** (**a**)

	Coefficients	SE	|t|	*p* Value
Intercept	0.3956	0.1138	3.477	0.0024
Stroop interference (score)	0.009227	0.002053	4.494	0.0002

**Table jcm-09-00609-t004b:** (**b**)

	Coefficients	SE	|t|	*p* Value
Intercept	71.28	6.039	11.8	< 0.0001
Systolic blood pressure (mmHg)	−0.1308	0.04482	2.919	0.0085

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
