# Peer review of "Risk Factors for Postural and Functional Balance Impairment in Patients with Chronic Obstructive Pulmonary Disease"

_jcm, 2020, doi:10.3390/jcm9020609_

Round 1

Reviewer 1 Report

The authors focus on an essential extrapulmonary manifestation of COPD, with a goal to identify risk factors for postural and functional balance impairment in a heterogeneous group of COPD patients. The article is overall well written and easy to read. However, several sections of the paper could be improved and specific comments are listed below.

Language, punctuations, etc.: Minor language mistakes are found throughout the manuscript, and the paper should be carefully proof-read before re-submitted.

E.g., Line 17-18: “However, risk factors underlying balance impairment in COPD have not fully established.” The word “been” is missing. Similar mistakes are seen also at other sections of the paper.

Introduction

Line 52-53: “Postural sway is impaired in COPD patients and associated with an elevated risk of falls [2].” Add information on how strong the association is.

Line 58-60: “These findings, as well as the fact that bronchitic patients experience more frequent falls than emphysematous patients [10], suggest that a specific COPD phenotype is at greater risk for impaired balance function”. In comparison to what?

Or should it be written:

“These findings, as well as the fact that bronchitic patients experience more frequent falls than emphysematous patients [10], suggest that specific COPD phenotypes could be at greater risk for impaired balance function.”

In general, the introduction is a bit unclear, going in different directions. It would be good if the authors could emphazise a bit more about what their study will add to what we currently know on the topic?  E.g., it is highlighted that “Factors and mechanisms underlying balance impairment in COPD patients have not been fully established” but it is not clear what this study adds in relation to what we currently know?

E.g., to which extent is measurement and/or the study sample, different from other studies?

Materials and Methods

Figure 1: Why were 9 controls excluded so late in the study? It is stated that “To match the age, sex, and BMI,” but then this could have been done before coming to the analysis?

Furthermore, how was the selection to exclude one control over another done? Were all remaining included controls matched identically with a COPD-patient on age, sex and BMI? It is essential that this is clarified.

For example, since p-values were borderline significant for postural sway (p=0.0496), it would be interesting to know how removing these 9 healthy controls affected the results? Especially considering that these 9 participants represented almost 30% of the total control sample.

Line 163-182: Statistical Analyses

How was the sample size determined in relation to the number of independent variables in the multiple linear regression model?

I suggest adding information on cut-offs to illustrate the strength of the correlations (see also comment in the discussion)

Results

Is it possible to combine Table 1 and Table 3 into one table? This, since all of these variables, represents the characteristics of the included participants.

The result section includes a considerable amount of data and different comparisons, could some be moved to an online appendix? I believe that this would increase the readability of the results section.

Line 194. Is the difference in BBS score (9.7%) correct?, I get it to 3%

Line 257-268: Is the use of the words: “up to” correct? For Table 4a, didn’t the model explain 66% of the variance in postural balance and for functional performance 83% of the variance? Why did you write “up to 66%” and “up to 83%”, respectively? P

Line 262. Change Table 4 to Table 4a

Line 263: Change Table 4 to Table 4b

Discussion

The discussion is written in a clear way with adequate comparison to previous studies. Considering the vast amounts of comparisons made between COPD and the controls in the results section, I would also recommend discussing these findings. Currently, this comparison is only mentioned once for one outcome. 

I am surprised that the authors do not discuss the possible impact of a relatively preserved balance in their included people with COPD and how this affects the interpretation of the results. For example, even if COPD and controls significantly differed on the BBS. The difference was small (3%), not clinically relevant and people with COPD scored 53 out of 56. Thus, their balance wasn’t really impaired.

Also, only AP sway velocity was impaired, and even though the difference was 29% ( I get it to 23%), this was borderline significant.  

The similar goes for the findings in the multiple regression analysis. How are these findings, in comparison to previous research? E.g., is there something that you measured that previous research have not that might explain your high explanation of the variation, especially considering the functional balance. Also, about the latter, even though functional balance was lower in COPD compared to controls. The mean score was 53 out of 56 on the BBS, indicating a preserved balance and how should we then interpret these findings?

Line 305: “Furthermore, the strong relationship between impaired postural sway and presence of diabetes” The correlation between diabetes and postural sway was r=0.456 which to my experience, often represents “moderate” relationships. However, if different cut-offs were used, provide this information in the methods section.

Reviewer 2 Report

General comments:

This is the study investigated the factors associated with impaired balance function in COPD patients compared to those in non-COPD patients and also investigated the relationships between balance function and several factors including body composition, muscle function, physical performance, neurocognitive function, plasma markers, and disease severity. They found that the AP mean sway velocity was higher (poorer postural balance) and the BBS was lower (poorer functional balance) in COPD than non-COPD. The postural balance in COPD was associated with presence of diabetes and fat mass, and the functional balance was associated with history of exacerbation, gait speed and transcutaneous oxygen saturation. These results in COPD are interesting, however, there are several concerns in this manuscript.

Major comments:                                          

Muscle function, physical performance and neurocognitive function were worse in COPD patients compared to control subjects (Table 3), which might induce the decreased balance functions (Table 2). Furthermore, 59% (20 out of 34) of COPD subjects were on long-term oxygen therapy. From these results, the physical activity, the strongest predictor of COPD mortality, was expected to be very low in recruited COPD patients compared to control subjects. Unexpectedly, however, PASE score (physical activity level) was not different between COPD and control (Table 1). This discrepancy is very difficult to understand. Author should discuss this discrepancy and explain the reason for non-reduced physical activity in COPD patients (e.g. the problem of matching process in control group etc.). Though postural balance was associated with presence of diabetes and fat mass by a multiple linear regression analysis (Table 4a), glucose concentration and fat mass in COPD were not different from those in control (Table 3). The high glucose concentration and high fat mass might be factors of impaired balance function not only in COPD patients but also in other subjects. Diabetes itself might influence the balance function as author describes in the discussion. Why diabetes influenced the balance function in only COPD but not in control? Though author described neurocognitive function and age are risk factor for postural balance function in the first sentence of discussion and in conclusion, these relationships were not statistically significant by multiple linear regression analysis (Table 4a). Author’s interpretation was misunderstanding and was not true. It seems that some factors, e.g. muscle function might associate with both balance functions, but the associated factors of postural balance and those of functional balance were completely different. Were these results observed only in COPD? Author should discuss about this point and referred the previous reports including other than COPD. P7, 235-242: Though the results of cut-off values in several factors were presented, the explanation how to define each cut-off value was not described. Author explained the cut-off analysis in the Method section.

Minor comments:

P7, L213-223: The associated factors with Postural Sway Velocity were presented, but the detail (AP, ML, AP-ML or Sway area) were not described. Author should explain the concrete method. P11, L347: As “Not all subjects conducted the BBS measurement”, author should describe the number of patients which performed BBS measurement in the Results section. P9, L262: “Table 4” should be “Table 4a”. P9. L263: “Table 4” should be “Table 4b”.

Round 2

Reviewer 2 Report

The suggested points were well modified.